# Exploring How Intent of Self-Harm Affects Trainee Healthcare Practitioners’ Views, Attitudes and Clinical Decision-Making in Northwest England: A Qualitative Study

**DOI:** 10.3390/ijerph22101563

**Published:** 2025-10-14

**Authors:** Destiny Priya Kumari, Kathryn Jane Gardner, Peter Taylor

**Affiliations:** 1Division of Psychology and Mental Health, School of Health Sciences, University of Manchester, Manchester Academic Health Sciences Centre, Manchester M13 9PL, UK; peter.taylor-2@manchester.ac.uk; 2School of Psychology and Computer Science, University of Lancashire, Preston PR1 2HE, UK; kjgardner@uclan.ac.uk

**Keywords:** self-harm, suicidal, non-suicidal, function, stigma, healthcare professionals, trainee, thematic analysis

## Abstract

Self-harm is often stigmatised by healthcare professionals. Little is known about how certain characteristics of the behaviour, like the degree of suicidal intent, affect clinicians’ judgements and responses. This study explored how intent of self-harm (suicidal or non-suicidal) affects trainee practitioners’ views and attitudes towards behaviour and clinical decision-making. A qualitative study using semi-structured online interviews was conducted. Interviews were audio-recorded, transcribed verbatim and analysed using a reflexive thematic analysis. Fifteen trainee healthcare practitioners (psychological wellbeing practitioners, clinical/counselling psychologists, nurses, and medics) were interviewed. Three themes were identified: (1) interpreting intent in self-harm: clinical utility and challenges, (2) the perceived responsibility of self-harm, (3) trainees’ struggle for equal care in a hierarchical system. Participants observed staff expressing pejorative views towards both forms of self-harm but did not share them. Across most clinical decisions, intent informed trainees’ judgements and beliefs. Clients presenting with suicidal-related self-harm received more urgent care but fewer therapeutic interventions. Trainees felt ambivalent about distinguishing intent. Nevertheless, this distinction was evident across treatment, risk and care decisions. A consistent approach towards suicidal and non-suicidal self-harm is important. Trainees should be supported in addressing difficult emotions arising from this work, helping to shift the blame culture and promoting a more empathic approach.

## 1. Introduction

Self-harm refers to intentional self-poisoning or injury to oneself, irrespective of the apparent purpose of the act, and includes a variety of behaviours like self-cutting, burning and overdoses [1]. Self-harm is a global health issue, affecting approximately 15 million people annually [2]. In 2014, 6.4% of individuals reported self-harming within their lifetime with 20.6% of these experiencing suicidal ideation [3]. These rates have significantly increased in 2023–2024 with the prevalence of self-harm rising to 10.3% and suicidal ideation rising to 25.2% [4]. Despite its prevalence, self-harm is often misunderstood and highly stigmatised. Public stigma involves labelling, stereotyping and discriminating against others based on perceived negative characteristics [5]. Common pejorative stereotypes include self-harm being selfish, manipulative and attention-seeking [6,7,8]. Healthcare professionals are not immune from societal stigma. In a Canadian sample of staff who worked in senior leadership in the area of client safety or mental health, 79% recalled personal experiences of discriminating against their clients and 53% observed colleagues’ discriminatory behaviour [9]. Similar negative attitudes have been found in UK healthcare settings, with clients labelled as difficult to manage, lacking co-operation and being untrustworthy [10,11,12,13]. Emergency department nurses viewed clients who attended accident and emergency (A&E) because of self-harm as less deserving of a bed compared with those who did not engage in self-harm [14].

Stigmatising views held by healthcare professionals may hinder client engagement. Individuals who self-harm often report feeling misunderstood or disregarded due to the lack of empathy and engagement from healthcare professionals [15,16]. Stigma can influence clinical decision-making, with some healthcare professionals either refusing treatment [17,18,19,20,21] or focusing solely on treating the physical injury whilst neglecting the mental health needs [22]. Additionally, stigma can discourage help-seeking behaviours due to a fear of judgement and rejection and a lack of trust and empathy in healthcare providers [23,24,25]. Those currently undergoing training as health professionals are the next generation of practitioners who will shape how care for self-harm is implemented in the future. Understanding trainee healthcare practitioners’ perspectives is essential, as reflecting on these views early in their training can help address biases and improve future practice.

Self-harm is not formally distinguished by intent in the UK NICE guidelines [1], which encompass behaviours with and without suicidal intent (e.g., suicidal and non-suicidal) [26]. This contrasts with a portion of the research literature that draws a distinction between non-suicidal self-injury (NSSI) and suicidal behaviour. NSSI involves intentional injury to the body without the intent to end one’s life and can serve a variety of functions including coping with emotional distress [27,28]. A suicide attempt (SA), in contrast, refers to injuring oneself with the intent to die [26]. However, research indicates that individuals can engage in self-harm and experience suicidal thoughts without having the intent to end their life [29,30]. The intention of self-harm is theorised to vary across a continuum [31]. Yet, despite this complexity, research has often treated NSSI and suicidal behaviour as distinct constructs, potentially oversimplifying the diverse nature of self-harm and suicidal behaviours [32].

There is debate about the classification of self-harm by intent in clinical practice [33]. It may be difficult for healthcare professionals to ascertain when intent has changed as clients may be unaware [34] or clinicians’ perception of suicidal intent may differ from the clients [35]. Physical healthcare professionals, focused on treating the physical injury, may not benefit from distinguishing intent, as their primary concern is managing the immediate physical harm rather than the underlying psychological motivation [36]. Alternatively, intent may add value in further understanding the behaviour and implementing focused interventions [37]. For example, several authors suggest that NSSI differs in motivations, functions and epidemiology from suicide attempts; thus, distinguishing the intent of self-harm behaviours is essential for fully understanding the behaviour and enabling effective treatment [38,39].

Clinicians’ perceptions of intent may affect the beliefs they form about instances of self-harm, including the occurrence of stigma [40]. Corrigan’s (2003) attribution model shows how beliefs around controllability predict beliefs about how responsible a person is for their mental health difficulties [41]. Unfavourable behaviours perceived as more controllable are more likely to attract stigma [42]. Studies on the perceived controllability of suicidal and self-harming behaviours have mixed findings. Gale et al. [43] found that mental health professionals viewed suicide attempts as less controllable and less predictable than conditions like anxiety or depression. Senf, Maiwurm, and Fettel [44] reported that healthcare professionals working with cancer clients often saw suicidality as a rational response to suffering, rather than impulsive. Yue et al. [45] found that psychiatric nurses perceived NSSI as difficult to manage, suggesting limited controllability, though repeated acts were sometimes viewed as deliberate or attention-seeking. The perception that someone has control over their self-harm and is choosing to do this can contribute to the stigma surrounding this behaviour [46], but it is unclear whether perceptions of intent moderate these judgements of controllability.

Whilst there is sufficient research exploring healthcare professionals’ views of self-harm as a whole and NSSI and SA separately, less is known about (a) how intent affects these views, (b) how these views influence clinical decision-making [47]. Understanding trainee healthcare practitioners’ perspectives is essential, as reflecting on these views early in their training can help address biases and improve future practice. The overall aim of this study was to qualitatively explore trainee healthcare practitioners’ experiences and understandings of self-harm to develop insight into where stigma is attached. The primary aim was to explore how the perceived intentionality of self-harm affects trainee practitioners’ views and attitudes towards the behaviour. A secondary aim was to assess how these attitudes influenced their treatment and clinical decision-making regarding the client’s care.

## 2. Materials and Methods

### 2.1. Design

A cross-sectional qualitative design was adopted using one-to-one interviews. A qualitative approach is well suited to capturing rich, contextualised insights that may not be accessible through structured quantitative measures alone, enabling the exploration of complex social and psychological phenomena (perceptions of self-harm). This study was part of a larger qualitative study of trainee practitioner attitudes on self-harm.

### 2.2. Participants

Participants were recruited through adverts distributed digitally to relevant university programme cohorts and paper adverts placed in relevant buildings on campus at universities in northwest England. Inclusion criteria included being aged 18 years or above and being a ‘trainee healthcare practitioner’. Trainee healthcare practitioner status was defined as studying on a university programme which would enable students to work in a healthcare practitioner role (e.g., trainee clinical psychologists, nursing associates, and medical students), and which included a healthcare placement or experience in clinical settings as part of the programme. Exclusion criteria stated that individuals who do not understand verbal and written English could not participate in the study. The target sample size was guided by information power [48]. Based on the focus of the study, twenty participants were deemed adequate to provide rich, in-depth insights into trainee practitioners’ attitudes and provide participant diversity. However, only 15 participants were recruited. As little new information emerged towards the end of data collection, it is unlikely that the additional five participants would have altered the findings substantially. However, it is important to acknowledge that this may have limited the breadth of perspectives captured in this paper and should be considered when interpreting the results. This study received approval from a university research ethics committee.

### 2.3. Measurements

A demographics form captured information regarding the participants’ age, university course, ethnicity, gender and lived experience of self-harm. The semi-structured interview schedule contained questions exploring participants’ conceptualisation of self-harm, their experiences of working with individuals who self-harm, the ways in which intention (defined dichotomously as suicidal or non-suicidal) affected their views and clinical decision-making, and the training and education they had received around self-harm behaviours. Example questions included the following: “What does it feel like as a trainee healthcare practitioner to have responsibility for the care of patients who self-harm with suicidal intent?”, “How does this differ from being responsible for the care of a patient who self-harms without suicidal intentions?”, and “How do you think your clinical decisions might differ for someone who self-harms with non-suicidal intentions compared to suicidal intentions?” The researcher reviewed interview schedules in the existing literature around the intent of self-harm and stigma to identify topics to discuss which align with the aims of the study [49,50,51]. While the interview followed a set structure, there was room for flexibility to follow up particular responses and comments [52]. The interview schedule was developed collaboratively with a linked study examining how the location of self-harm influences trainees’ views and clinical decision-making. As such, the schedule included questions relevant to both studies; however, only data relating to the present study was analysed. A copy of the interview schedule is available in Appendix A

### 2.4. Procedure

Individuals interested in participating could contact the researcher. Potential participants were invited to an initial meeting with the researcher, either in-person at the university or online via video call. All the participants chose online meetings. Informed consent was obtained first. The demographics form was sent to participants to complete prior to the interview. The qualitative interview lasted approximately 60 min and was guided by an interview schedule of semi-structured, open-ended questions. All participants were reimbursed for their participation.

### 2.5. Analysis

Interviews were transcribed verbatim by DK. Reflexive thematic analysis (RTA) was used to analyse the data. RTA is a theoretically flexible process where the researcher explores and develops patterns of meanings across a dataset with the aim of creating a logical, consistent interpretation of the data, established through the participant’s voice [53]. Braun & Clarke’s [53] RTA entails six stages which include data familiarisation, code generation, theme generation, reviewing themes, defining themes and reporting the results of the analysis through conducting a write-up. Inductive RTA was used through creating descriptive codes which summarised the essence of participants’ narratives. Codes were refined to ensure each excerpt represented one code. The analysis focused on semantic and latent levels of analysis through identifying the explicit and deeper, implicit meanings behind participants’ expressions. This study used a critical realist framework which assumed that a shared external reality can be inferred through the research, but that the researcher’s perspective would affect how inferences could be made about this reality [54]. See Appendix A for further details of the analysis.

### 2.6. Reflexivity and Trustworthiness

The researcher was a 22-year-old, British Asian, MSc Clinical and Health Psychology student with personal experience of self-harm. They have worked across both private and public mental healthcare settings both in the community and in client settings as a support worker and an assistant psychologist where self-harm behaviours are prevalent. Author 2 (PT) is a Clinical Reader and Clinical Psychologist with expertise in qualitative research and clinical practice, contributing to the analysis and reflexive discussion as a supervisor. Author 3 (KG) is a Reader in Clinical Psychology with a research interest in understanding self-harm and its functions and mechanisms. The findings should be read in this context.

To ensure the trustworthiness of the findings, the research team followed three principles to ensure the analysis was robust [55]. Credibility refers to the ability to ensure that findings are accurate and are based on having a solid understanding of the data. To ensure this, data were transcribed verbatim so direct quotes could be evidenced to support the themes without participants’ voices being lost through the interpretation. Data were analysed by one researcher. However, supervision was utilised with a clinical psychologist in the research team to discuss the patterns which were arising and any discrepancies. Dependability refers to the ability to replicate the same research process. To ensure dependability, the researcher consistently followed Braun and Clarke’s [53] reflexive thematic analysis process. Transferability refers to the degree to which study findings can be generalised. To ensure judgements could be made about the degree of transferability of the findings, participant characteristics and a clear description of the context were provided. This was used to consider the potential impact on the findings.

## 3. Results

### 3.1. Participant Characteristics

The interviews were conducted remotely with a sample of all-white trainee healthcare practitioners (*N* = 15) working within National Health Service (NHS) teams as part of their placement. The mean age was 29 years (range = 20–47). Most participants were female and from psychology-based professions (*N* = 12, e.g., trainee clinical psychologists, trainee counselling psychologists, and psychological wellbeing practitioners) and personally knew somebody who engaged in self-harm. Half of the sample (*N* = 8, 53%) had lived experience of self-harm. Participant demographics are summarised in Table 1.

### 3.2. Thematic Analysis

Three themes emerged through the inductive thematic analysis: (1) interpreting intent in self-harm: clinical utility and challenges, (2) the perceived responsibility of self-harm, (3) trainees’ struggle for equal care in a hierarchical system (see Figure 1). An overarching, cross-cutting theme was an inconsistency between how participants felt that intent was an unhelpful guide for informing their clinical practice but simultaneously gave examples of making clinical decisions based on perceptions of intent.

#### 3.2.1. Interpreting Intent in Self-Harm: Clinical Utility and Challenges

There were inconsistencies within and across transcripts regarding the utility of distinguishing intent within clinical practice. Trainees demonstrated ambivalence, sometimes using intent to guide care decisions whilst at other times criticising this approach. Moreover, trainee practitioners commonly assessed intent using factors such as methods, severity and location of self-harm, while clients’ self-reported intent was sometimes questioned, with concerns about their authenticity. This highlights the complexity of assessing and categorising intent with significant implications for treatment planning and training.

##### Conflicting Views of Intent in Clinical Decision-Making and Training

Trainees’ beliefs around self-harm intent differed from their clinical practice. Whilst most trainees supported equal care for all clients, irrespective of intent, they acknowledged that intent shaped clinical decisions and treatment dynamics. Specifically, intent influenced risk appraisals and service eligibility, with NSSI being seen as lower risk, and SA being seen as higher risk: “I think I would probably take it to be more severe if there was suicidality, like a more urgent issue if there was immediate suicidality” (Medicine 2). Some participants recognised the utility of intent in determining eligibility for service referrals, with NSSI typically being treated in primary care settings and within the community. For example, DClinPsy 4 stated “…more likely to work with them say in the primary care setting…” In contrast, suicidal behaviour was considered more appropriate for secondary care such as mental health inpatient services. However, others noted that due to under-resourcing, services often triage referrals without considering intent, leaving those working in low-level services—such as primary care mental health teams and general practitioners—feeling unprepared to manage suicidal self-harm. For example, Trainee Associate Practitioner 1 stated “…we were supposed to be very low-level risk, in the community… but the majority of the people we work with do you have suicidal intent…”. Mental Health Nurse 3 supported this by highlighting how community services tend to feel helpless because they offer interventions which they feel are inappropriate for those engaging in suicidal self-harm “…they [those engaging with suicidal self-harm] wouldn’t be suitable for us and we would need to refer them on… I personally would feel responsible to get them help and not sure where to look.” Participants suggested that grouping self-harm by intent could be helpful as a process for learning different approaches and presentations and for the dissemination of client information: “…in practice it might be useful for the trainees to understand the differences in like approaches and techniques and skills …” (DCounPsy 1).

Some participants argued that labelling intent was problematic for clinical decision-making, as it oversimplified the fluid nature of intent and ignored the risks related to NSSI and passive suicidality such as accidental suicide. For example, Medicine 2 stated “…when people are very much on in the grey areas…passive suicidal ideation… it might be difficult to know which box to put them in…”. Participants described the tension between simplifying the phenomenon and the stigmatisation of clients due to labelling—“…we don’t like putting people in boxes… I wouldn’t want that to be obvious to clients because I, don’t like people to feel different because of intent…” (DCounPsy 1). Both clinical and counselling psychology participants were particularly critical of the binary approach, as it excluded clients who did not fit this dichotomy, emphasising favour for the continuum view of self-harm. For example, Medic 2 stated “…if someone isn’t suicidal… that could escalate into becoming suicidal, so I think it’s, the two categories sometimes there is a bit of an overlap, and it isn’t just as black and white…”.

The presence of suicidal intent also guided the consideration of treatment options. Where suicidal intent was present, there was a greater focus on immediate risk management and safety. For example, DClinPsy 1 stated “…people’s feelings can be neglected when there is suicidal intent… managing that risk but they almost forget to stop and understand what’s going on [for] the person…” There was a greater sense of urgency associated with suicidal intent, and at times more restrictive practice such as increased observation levels, removing bathroom privacy and removing access to leave off the ward were supported. For example, Trainee Associate Practitioner 2 stated “I think you could feel a bit more panic if someone is saying that they’re acutely suicidal…more of a sense of urgency… make sure that they’re in a safe, contained environment and they stop…” Where non-suicidal intent was present, there was a greater focus on understanding the person holistically and encouraging autonomy through promoting the self-management of NSSI within community settings. For example, DClinPsy 3 stated “… if somebody with suicidal intent, there’s an acuity there and you respond to the risk whereas… somebody without it, is more like you’re thinking of a longer-term kind of response.” Although trainees perceived intent as a fluctuating factor, the perception of having more time to work with NSSI suggests that at times, the same clinicians viewed intent as fixed.

Participants from psychological backgrounds more often discussed intent being relevant to treatment decisions because psychological approaches tend to focus on understanding the underlying emotional and cognitive factors which drive self-harm behaviours. For example, DClinPsy 4 stated how psychological professions may have “…more time to explore what that serves for them, why they’re engaging in it and what kind of function does it have for them.” Some psychology trainees advocated for a person-centred approach. They criticised the binary nature of intent as it excluded treatment decisions for clients who did not fit this dichotomy, emphasising favour for the continuum view of self-harm which acknowledges that the intent behind self-harm may vary in degree and nature over categorical approaches (e.g., categorising intent as absent or present). In contrast, participants from medical backgrounds felt intent was seen as less relevant and not helpful for making clinical decisions. Some participants from medical backgrounds explained that this was because intent plays a less direct role in treatment options which are often diagnosis-driven and generic (e.g., medications) and focus on managing symptoms and stabilising an individual. For example, Medicine 3 stated “in a psychiatric sense it’s kind of you know what drugs are for what conditions and you prescribe them…” Mental Health Nursing 1 also stated that intent was meaningless for nursing professions because “…regardless of what the intent is…all self-harm can have the same implications or consequences like killing yourself…”.

##### Client vs. Trainee Judgements of Intent

Trainees often dismissed or questioned clients’ own judgement of intent. This was particularly apparent during cases where clients verbalised suicidal intent. Some trainees reported that they and their colleagues sometimes questioned the authenticity of verbalised suicidal intent, perceiving it as disingenuous. Some participants raised doubts that some instances of disclosing intent were accurate. For example, Mental Health Nursing 1 stated “I feel in some instances… They’re saying it [disclosing suicidality] because they know it will create a certain response, but it is not necessarily something that they actually intend to do.” DClinPsy 5 supported this when they said “… you feel like you’ve had a conversation [about SA] … coming away thinking like, don’t believe you.” However, a few trainees advocated for the acceptance and validation of clients’ experiences despite clinicians’ uncertainties, acknowledging the difficulty of disclosing intent to healthcare professionals. For example, DClinCoun1 stated “…I think it’s very much about you know viewing the person as a person no matter what they’re going through, no matter what their adverse experiences or you know no matter what they’ve chosen to do…”.

Clinical characteristics such as the method, location and severity of self-harm emerged as key factors that trainees used to assess clients’ intent. For example, methods associated with more physical risk like cutting near a major blood vessel were linked to more suicidal intent. DClinPsy 2 stated “the risk associated with a ligature is obviously very difficult and high and it can quite easily lead to loss of life even if it’s not intentional… cutting may be less so suicidal intent…” Although not explicitly asked, trainees referenced these factors, suggesting they played a significant role in judging the severity and intent of self-harm. This indicates that clinical characteristics influenced their decision-making, even without direct inquiry.

##### Preference vs. Priority: NSSI and Suicide Risk

Clinicians reported paradoxical statements when commenting on their optimism and responsiveness to care based on perceived intent. Trainees evidenced a preference to work with NSSI compared with SA, viewing these individuals as more amenable to treatment (e.g., higher motivation, increased engagement with staff). For instance, Mental Health Nursing 1 stated clients with non-suicidal intent “…their motivation to find alternative methods or to follow like a care plan… I feel as though their motivation is higher.” Despite less optimism seen in trainees when treating suicidal individuals, an ethical obligation to prioritise their care was echoed by most, leading to increased responsiveness to care. Some trainees shared less optimistic views of recovery when suicidal intent was present, suggesting that persistent suicidal feelings were discouraging and indicative of limited progress. For instance, Medicine 1 stated “If someone’s continuing not to feel suicidal…you’re glad that they are not becoming any worse or deteriorating…if they’re continuously feeling suicidal, I suppose that does be disappointing and deflating in itself…” However, the same trainees remained responsive to these clients, perceiving them to require more urgent care than those with non-suicidal intentions. For example, DClinPsy 2 stated that “…in one circumstance there’s a risk to life…that always has to be a priority.”

#### 3.2.2. Perceived Responsibility of Self-Harm

Trainees assigned different levels of responsibility for self-harm and seeking treatment based on intent. The responsibility shifted from the client to the healthcare professional and then to the healthcare system. Their appraisals determined how they worked with individuals and their clinical decisions.

##### Responsibility for Behaviour

A subset of trainees talked about suicidal behaviours in a manner suggestive of a sense of the client having more control over their actions than those who engaged in NSSI. For example, Nursing Associate Practitioner 1 stated “…you may see people who…ligature with ligatures they can untie or people who restrict their diet but not to a point where you know they’ll still eat a little bit, or they’ll still drink a little bit…” This suggests that some trainees viewed SA as a purposeful, planned act, implying clients may exercise some control over their actions. It is important to note that some participants had experience working in inpatient settings, which likely influenced their narratives. Witnessing repeated ligature incidents on a ward, for example, provides a stark contrast to the lay perception of a suicide attempt or the situations encountered in community secondary care services, potentially shaping their understanding and judgement of self-harm and suicidal intent.

NSSI was viewed by some as an impulsive act, suggesting less control. For example, Trainee Associate Psychological Practitioner 1 stated “…if they [non-suicidal clients] don’t do it [self-harm] frequently sometimes they experience a loss of control and they don’t really remember why they’ve done it, and they can’t explain why they’ve done it…” However, other participants believed clients, to some extent, have self-awareness of the risks associated with NSSI and are making informed choices. DClinPsy 1 said “…I’ve spoken to people who’ve… been very clear on kind of where they would and would not cut and like aware of, I guess the risks associated…”. Some trainees recognised that the controllability of self-harm could vary with mental health conditions, such as psychosis or mania, affecting a person’s control over their behaviour regardless of the intent. For example, DClinPsy 5 stated “…if someone was experiencing psychosis or mania and engaging in self-injurious behaviour, I would argue, they don’t have control, therefore they’re not responsible.”

##### Responsibility for Treatment

Despite sometimes viewing NSSI as less controllable, some trainees assigned more responsibility to clients engaging in NSSI for obtaining and engaging with treatment. For example, TAPP 2 suggested offering a “… safety sort of package that this person could access, so that if they were cutting in their room…they would be able to help themselves to some degree [for NSSI] …”. Moreover, trainees found it easier to maintain professional boundaries and manage situations with NSSI. For example, DClinPsy 2 stated that having a boundaried approach rather than an emotional response to NSSI helps the therapeutic relationship “… it allows me to have a little bit more erm, practicality in my approach… I think if I wasn’t desensitised, some things would really shock me and that might come across to the person and then they might be kind of put off about talking about it again…”.

In contrast, trainees felt a greater sense of responsibility for the outcome of treatment for suicidal clients, particularly in psychological professions. DCounPsy 1 stated “…it’s hard to carry that weight of feeling like you have some responsibility but… we have no way of making that person not go and die by suicide…”. They reported taking a proactive, indirect role in caring for clients with suicidal intent. Interventions for SA focussed on risk management, following protocols and speaking to senior members of the team to share the risks, which may have led to suicidal clients taking a more passive role in their care compared with NSSI.

Heightened personal responsibility for clients with suicidal intent contributed to anxiety for some trainees. Trainees felt that they had an ethical, moral obligation to treat these individuals but also wanted to prevent harmful consequences to themselves that might result from the death by suicide of a client under their care. This included fears of being blamed or held responsible for the death of the client. For example, Mental Health Nursing 1 noted that “…at uni we are taught so much about things like coroners court or the thought of you could lose your pin…I don’t want to get involved in that…” This avoidant approach towards working with suicidality, increased feelings of inadequacy and helplessness with decision-making.

#### 3.2.3. Trainees Struggle for Equal Care in a Hierarchical System

Some trainees evidenced a drive for change, to see equal treatment for both forms of self-harm within the healthcare system. However, many trainees also highlighted that the clinical decisions they make can be influenced by system constraints and hierarchical structures, leading to a disconnect between the care they hoped to provide, and the actual care delivered. For example, when discussing how they wanted to provide more in-depth care for NSSI, Mental Health Nursing 2 stated “…they [the service] don’t have things to offer erm, and that people genuinely want help, and they’re turned away with a bath and a cup of tea.” A common thread in the transcripts was trainees’ acknowledgment that the healthcare system should bear some responsibility for the differential treatment of clients. They identified systemic issues such as inadequate funding, limited training, poor care pathways and bed shortages, as barriers to providing equal care. For example, Medic 3 stated “…it’s just that we can’t [offer equal care] because of staffing and because of, we can’t give them as much time because like client safety is the priority…” Mental Health Nurse 3 also highlighted the challenges of getting support, irrespective of intent “…there’s so many reasons for that like the understaffing, like lack of funding, lack of beds like all time scales that you want people in and out…” Despite recognising this, trainees still felt obliged to hold that personal responsibility with suicidal intent, as seen in theme two.

Some trainees expressed a desire to challenge stigmatising views within their care teams, particularly from supervisors or senior staff. However, they often felt powerless due to their trainee status and the hierarchical structure of the healthcare system, which made it difficult to challenge negative attitudes. For example, Mental Health Nursing 1 stated “…I did speak up [to colleagues] and kind of say I don’t think it [NSSI] is attention-seeking…it was kind of shot down … then you just lose the ability to verbalise and because you are a student it’s almost like what you’re saying has less weight…”.

## 4. Discussion

This study explored trainee healthcare practitioners’ attitudes towards self-harm, and how perceived suicidal intent influences views and clinical decision-making. Three themes were identified: (1) interpreting intent in self-harm: clinical utility and challenges, (2) the perceived responsibility of self-harm, (3) trainees’ struggle for equal care in a hierarchical system. The participants observed but did not share the pejorative views about self-harm (both with and without suicidal intent) expressed by other staff members. Despite most trainees advocating for equal care, irrespective of the intent of self-harm, implicit ideas regarding intent appeared to influence trainees’ work with self-harm. This suggests that intent may have an unspoken role within clinical decision-making, leading to differential care being received for NSSI and SA. The participants showed a preference for working with NSSI over SA. However, their appraisals of SA being higher risk than NSSI prompted more urgent responses, prioritising immediate risk management, environmental safety and the prevention of future attempts. In contrast, trainees proposed more therapeutic, psychological interventions for clients who engaged in NSSI, aimed at working towards longer-term goals such as supporting re-integration into the community. This suggests that intent influenced the type and immediacy of care delivered, with SA receiving more immediate risk-focused intervention and NSSI receiving more planned, long-term therapeutic support.

Participants expressed ambivalence regarding the role of intent in clinical decision-making. While some clinicians saw value in understanding intent to guide treatment decisions, others questioned its relevance or practicality in determining appropriate care. This study found that trainees’ judgements of risk may be influenced by intent, with a higher risk associated with SA. However, some trainees noted the difficulty and potential inaccuracy of using intent alone to determine risk, especially when severe NSSI or passive suicidality were evident. This highlights the limitations of static risk judgements based solely on intent as it overlooks the dynamic and complex nature of self-harm behaviours [56]. It could lead to underestimations or overestimations of risk [57], potentially resulting in inadequate care or false reassurances [58]. Further investigation into how intent informs risk-related judgements is needed.

Trainees’ attitudes towards treatment differed according to the intent of self-harm. Non-suicidal behaviours were seen as more amenable to psychological interventions. Yet, it was unclear whether they were being delivered in practice as there was also a greater focus on self-management. This may contribute to some clients who engage in NSSI feeling dismissed by healthcare services [49,59]. On the contrary, the presence of suicidal intent often led to more passive, risk-averse care, with minimal client autonomy or collaboration. Although psychological approaches are recommended for managing suicidal behaviours [60,61], the continued emphasis on risk management strategies, such as objective risk assessments and safety plans [62,63], may influence trainees to prioritise risk management over therapeutic engagement. This aligned with the data where there was a tendency to associate inpatient care with suicidal intent, though it may also reflect the background and experiences of the sample, most of whom had prior inpatient placement experiences. Yet most people who self-harm, even with suicidal ideation, are outpatients [64]. This raises questions around how care may differ when suicidal intent is present within community settings.

This study revealed conflicts between trainees’ motivation to engage with clients based on self-harm intent. While some trainees expressed greater optimism and motivation to treat cases of NSSI, they often deprioritised cases where suicidal intent was not explicitly evident. This pattern reflects the existing literature, which states that clinicians often prioritise care based on the perceived severity of self-harm, with suicidal behaviours receiving more urgent attention [65]. For example, self-cutting behaviours, typically associated with NSSI, were perceived as less serious than overt suicidal behaviours and were less likely to receive psychosocial assessments or prioritised care [66,67], even though they may still indicate significant psychological distress [68,69].

Moreover, this study highlighted the disconnect between participants’ perception of intent and those expressed by the clients themselves, demonstrating the drawbacks of distinguishing intent. Clinicians often interpreted self-harm through their own lens, which may not align with how clients understand or convey their actions. This mirrors findings from Cahill & Rakow [70] and Kolochowski et al. [71], who emphasised that clinician bias and the variability of clinicians’ interpretations of intent can lead to inconsistencies in treatment decisions. Bewick, McBride & Barkham [72] suggested that misalignments may be partly due to objective assessments that fail to capture the nuances of client experiences. This highlights the challenges in aligning clinical assessments with clients’ motivations and questions the accuracy of clinicians’ assumptions in treatment decisions.

This study showed that intent influenced perceptions of control over self-harm behaviours, with SA viewed as more controllable than NSSI. However, contrary to attribution theory [41], which suggests that more controllable behaviours are more stigmatised, NSSI was not necessarily perceived more negatively. This may reflect the tendency to view self-harm as a behavioural choice rather than as a symptom of mental health, as seen with depression and anxiety [73,74]. The literature presents conflicting views on the controllability of both NSSI and SA, with some suggesting SA is a purposeful act, while others argue individuals are not in control of their actions [30,75,76,77,78,79].

Despite the perception of suicidal behaviours as being more controllable, trainees expressed greater urgency and support for clients exhibiting these behaviours. This reflects a broader fear of being held accountable for adverse outcomes, which often leads to a more passive risk-averse approach to care [80]. For example, clinicians felt like they needed external control over a client until they were no longer engaging in suicidal acts and could think rationally [81]. This study suggests that trainees’ responses to self-harm may be influenced by factors beyond perceived controllability, such as a fear of professional repercussions, worry about making mistakes and a heightened sense of responsibility for treatment outcomes—echoed by the previous literature [14,82,83,84].

Finally, this study revealed concerns around systematic and organisational factors, such as training gaps, resource limitations and hierarchical structures, which impact care delivery. These challenges have been well documented in the literature [85,86,87,88,89]. Addressing these issues is crucial for improving the consistency and quality of care, particularly in ensuring equitable treatment for all individuals, irrespective of intent.

This study has several limitations. First, it can be noted that our findings reflect trainees’ perceptions of the suitability of interventions rather than evidence of their actual delivery. Because trainees may have drawn on multiple experiences across different healthcare settings, the availability and feasibility of psychological interventions could differ significantly across these contexts. This contextual distinction should be considered when interpreting the findings. The use of an all-British white sample from northwest England may limit the transferability of the results. Non-white practitioners may hold different views; however, this cannot be assumed without further research. Similarly, the gender imbalance within the sample further reduces the representativeness of the sample. If such differences are present in practice, this could influence clinical decision-making in ways not captured by this study. This is important to consider, as non-white clinicians make up a considerable portion of the workforce [90]. Moreover, exploring this would contribute towards understanding if demographic factors such as ethnicity and gender influence views, attitudes and clinical decision-making. Similarly, while over half the sample reported lived experience of self-harm, no clear differences emerged between participants with and without lived experience, though further work could explore this in more depth. A self-selection bias could have skewed the findings, as more than half of the participants had personal experiences with self-harm. Additionally, the sample, although mixed, consisted mainly of psychologists or medical professions with an interest in psychology. This may have influenced the views of intent compared with what may have been found in medical or nursing professionals without these biases. Future research should compare views and attitudes across the profession. It was challenging to distinguish between trainee’s own views and their perceptions of colleagues’ views. Social desirability bias may have led participants to mask their views, attributing them to colleagues instead. As trainees enter the healthcare system, they may be affected by colleagues and pressure to behave a certain way to fit in the group [91]. The impact of colleagues’ influence should be explored further. The discussion of intent was often shaped by the way interview questions were framed, with the interviewer typically presenting intent as a fixed attribute of the individual (e.g., whether the person has intent or not). This framing may have encouraged participants to view intent as a static quality of the person, rather than considering it as a dynamic and contextual feature of self-harming behaviour that can exist along a continuum.

These results suggest several educational and clinical implications. Currently, national guidance states a way of working with self-harm which does not make a distinction using intent [1]. However, intent had an unspoken influence on trainees’ clinical judgements, causing confusion and conflict. This could lead to issues in clinical practice such as inconsistent care and a lack of transparency in decision-making. As trainees may sometimes be led by their own underlying beliefs and assumptions rather than standardised protocols when making care decisions, the awareness and use of self-harm guidelines remain important to ensure a consistent response to care [92]. At the same time, guidelines emphasise the importance of a dynamic, individualised, needs-based approach [93] supported by risk formulations [94]. Strengthening training needs to address how to balance consistent care with flexibility and may help practitioners to consider intent in a transparent way, without undermining evidence-based practice.

There is a clear need for more focused training around intent and how clinicians should intervene, as specialised self-harm training is currently lacking [95,96,97]. Standardising training across the UK could help promote more consistent practice. Intent was addressed to varying degrees across universities with a heavy focus on suicidality, resulting in disparate levels of knowledge and attitudes towards self-harm. There is an unclear relationship between increased education and less stigma towards self-harm [68,98,99]. This inconsistency may stem from training content being overly general or focused on broader mental health issues, making comparisons between training courses challenging. Tailoring training to address specific forms of self-harm could help reduce stigma and improve care provision. Anti-stigma training has been shown to enhance clinician attitudes and increase help-seeking behaviours among clients [100,101]. However, challenges such as limited staff time and financial resources must also be considered to ensure access to adequate training.

Ongoing support for trainees, such as clinical supervision, reflective practice and Schwartz rounds, are crucial [100]. Schwartz rounds can be defined as meetings with healthcare staff which aim to reflect on the emotional impact of delivering patient care to support staff wellbeing and improve teamwork [102]. This would support trainees and clinicians to better recognise how their own beliefs and assumptions can affect their clinical decision-making and help them to feel adequately trained and supported. Additionally, a culture shift towards a “responsibility without blame” approach [102,103], which acknowledges errors made in healthcare within the context of the challenging environments healthcare professional’s work, could facilitate learning and improve practice. Shared decision-making models may further enhance collaborative working and positive risk-taking and reduce clinician burden [104,105]. Therefore, the impact of contextual factors on clinical decision-making requires further consideration. Systemic factors may explain the inconsistency between national guidance produced for “perfect care” and the imperfect environment in which staff are operating.

## 5. Conclusions

To summarise, this study is one of the first to explore how perceptions of intent influence the views and decision-making of trainee clinicians regarding self-harm. Trainees did not express overtly negative views towards NSSI and SA, but implicit ideas about intent led to the belief that NSSI and SA required different treatment strategies and were viewed differently around the perceived responsibility which clients held for different forms of self-harm behaviours. These unspoken ideas, assumptions and views around NSSI and SA are not captured in the guidelines but still informed care decisions. It is important to highlight this, as such assumptions may result in a lack of transparency and consistency in decision-making which could be problematic. A standardised approach to defining and treating both forms of self-harm is needed, achievable through reviewing evidence-based guidelines, improved training and providing support for trainees.

## Figures and Tables

**Figure 1 ijerph-22-01563-f001:**
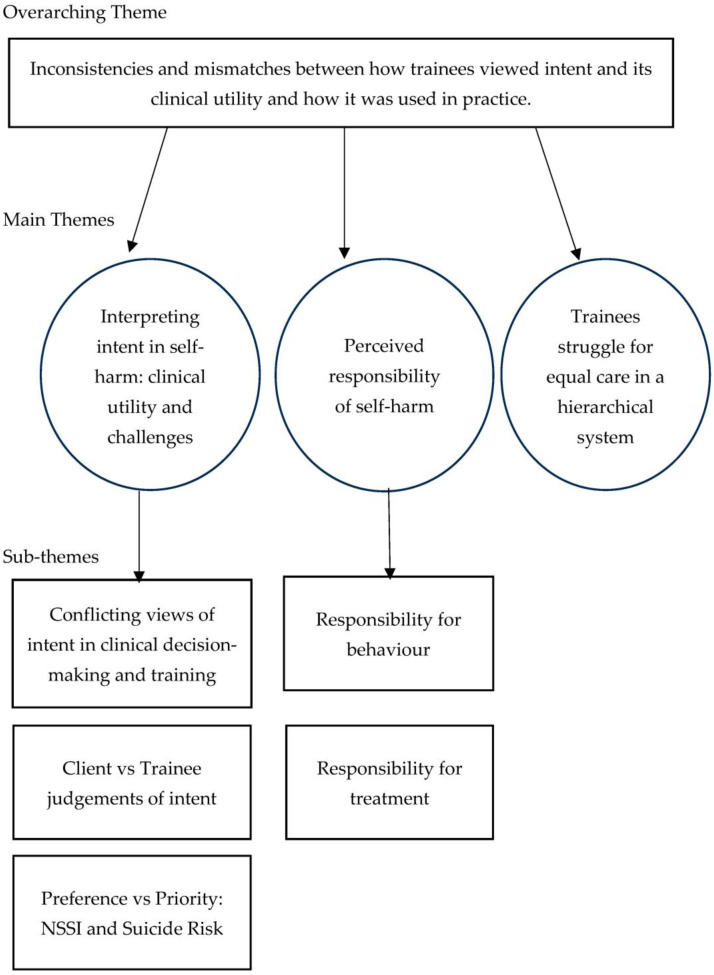
Thematic map illustrating themes and codes.

**Table 1 ijerph-22-01563-t001:** Summary of participant characteristics.

Characteristic	*N*
Age M ± (SD)	29.2 ± (8.02)
Gender	
Female	14
Non-binary	1
Ethnicity	
White	15
Programme	
DClinPsy ^1^	5
Mental Health Nursing	3
Medicine	3
TAPP ^2^	2
DCounPsy ^3^	1
NAP ^4^	1
Lived experience of self-harm	
Yes	8
No	7
Personally know someone who has self-harmed	
Yes, close family/friend	12
Yes, distant family/friend	2
No	1

^1^ Doctorate of Clinical Psychology, ^2^ Trainee Associate Psychological Practitioner, ^3^ Doctorate of Counselling Psychology, ^4^ Nursing Associate Practitioner.

## Data Availability

Given concerns around confidentiality, the data that supported the findings of this study are not available from the corresponding author.

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
