# Peer review of "Exploring How Intent of Self-Harm Affects Trainee Healthcare Practitioners’ Views, Attitudes and Clinical Decision-Making in Northwest England: A Qualitative Study"

_ijerph, 2025, doi:10.3390/ijerph22101563_

Round 1

Reviewer 1 Report (Previous Reviewer 1)

Comments and Suggestions for Authors

Thanks to the authors for their thoughtful responses to all feedback points and concerns, I'm happy for the manuscript to be published in this form

Author Response

No comments and suggestions from Reviewer 1#

Reviewer 2 Report (New Reviewer)

Comments and Suggestions for Authors

Dear Authors,

Thank you for providing us with an interesting and relevant paper. The topic is important and certainly warrants close attention.

The manuscript does not present major methodological or formal issues. However, I believe there is a substantial gap in the discussion that should at least be acknowledged.

Already in the introduction, the statement “Common pejorative stereotypes include self-harm being selfish, manipulative and attention-seeking [6–8]” risks oversimplifying the issue. While it is true that many clinicians fall into this trap, it is also clinically inaccurate to suggest that patients who self-harm, particularly those engaging in NSSI, are never manipulative. This omission likely stems from the paper’s approach of treating NSSI and SA in isolation, without considering the diagnostic and nosological context. NSSI is disproportionately more common in individuals with personality disorders, where manipulative behaviors are also prevalent. Conversely, SA in patients with severe depression or psychosis carries a much higher level of risk and urgency. While I agree that NSSI is often under-recognized, the current discussion risks implying an equivalence in severity between NSSI and SA, which is problematic.

Another concern relates to the following passage:
“Trainees’ attitudes towards treatment differed according to the intent of self-harm. Non-suicidal behaviours were seen as more amenable to psychological interventions. Yet, it was unclear whether they were being delivered in practice as there was also a greater focus on self-management.”
If, as I understand, the focus of the paper is on acute hospital care, how could psychological interventions realistically be delivered in this setting? Clarification here would be helpful.

Finally, although limitations such as ethnicity are discussed, the significant gender imbalance in the sample is not addressed. With most participants being female (14/1, plus 1 non-binary), this represents a major limitation and should be explicitly acknowledged.

Additionally, Figure 1 appears blurred and should be corrected.

Author Response

Comments and Suggestions for Author 3

  1. The manuscript does not present major methodological or formal issues. However, I believe there is a substantial gap in the discussion that should at least be acknowledged. Already in the introduction, the statement “Common pejorative stereotypes include self-harm being selfish, manipulative and attention-seeking [6–8]” risks oversimplifying the issue. While it is true that many clinicians fall into this trap, it is also clinically inaccurate to suggest that patients who self-harm, particularly those engaging in NSSI, are never manipulative. This omission likely stems from the paper’s approach of treating NSSI and SA in isolation, without considering the diagnostic and nosological context. NSSI is disproportionately more common in individuals with personality disorders, where manipulative behaviours are also prevalent. Conversely, SA in patients with severe depression or psychosis carries a much higher level of risk and urgency. While I agree that NSSI is often under-recognized, the current discussion risks implying an equivalence in severity between NSSI and SA, which is problematic.

We thank the reviewer for this thoughtful feedback. We agree that language around self-harm and it being a ‘manipulative act’ has historically been problematic and can risk reinforcing stigma, particularly for individuals with a mental health diagnosis.

Our intention in citing stereotypes like ‘selfish’, ‘manipulative’ or ‘attention-seeking’ was to illustrate pejorative perceptions reported in the literature and our sample (or their work teams) rather than to endorse them. We also recognise the importance of framing non-suicidal and suicidal self-harm within their diagnostic and clinical contexts and that risks/presentations can differ substantially between these groups.

We wish to:

  1. Clarify that we are reporting stereotypes, not supporting them.
  2. Highlight that current UK practice and professional guidance actively discourages framing self-harm as ‘manipulative’ (e.g. NICE’s self-harm guidelines NG225 (2022), the NHS Self-harm Competency Framework (2022), ethical codes of the BPS and HCPC). Importantly, such guidance likewise cautions against framing individuals with a diagnosis of borderline personality disorder (BPD) as ‘manipulative’ – recognising that this label is outdated, stigmatising and inconsistent with contemporary, trauma-informed and person-centred practice.
  3. Acknowledge that while diagnostic differences (e.g. mental health diagnoses) may influence the presentation and trainees’ views/attitudes/clinical decision-making towards self-harm, our study did not stratify participants’ attitudes according to the client/patient’s diagnosis.
  4. Note that whilst NSSI is not uncommon amongst people diagnosed with borderline personality disorder it is now recognised that self-injury also occurs widely in other populations and is by no means particular to those with borderline personality disorder (e.g. Butler & Malone, 2018; https://doi.org/10.1192/bjp.bp.112.113506 ; Victor & Klonsky, 2014; https://doi.org/10.1016/j.cpr.2014.03.005 ; Bentley et al., 2014; https://doi.org/10.1177/2167702613514563).

We hope that these points have clarified that our approach aligns with current UK clinical practice, which is moving away from pathologizing or stigmatising language and towards understanding self-harm behaviours contextually and compassionately.

We agree that it is vital not to imply an equivalence in severity between NSSI and SA. We would like to clarify that our manuscript does not suggest NSSI and SA carry the same level of risk. The difference in perceived risk associated with these behaviours is one of the findings highlighted in the study. Our findings show that trainees consistently perceived SA as carrying a greater immediate risk and urgency than NSSI, and we have included multiple examples illustrating this in the results and discussion section. We also note that different forms of self-harm prompt different clinical responses, with SA triggering more risk-focussed interventions and NSSI prompting longer-term therapeutic approaches.

Please see below for explicit references to differences in level of risk for NSSI and SA:

Page 7, Line 276 – “. Specifically, intent influenced risk appraisals and service eligibility, with NSSI being seen as lower risk, and SA being seen as higher risk. “I think I would probably take it to be more severe if there was suicidality, like a more urgent issue if there was immediate suicidality” (Medicine 2).”

Page 8, Line 316 – “There was a greater sense of urgency associated with suicidal intent… Where non-suicidal intent was present, there was a greater focus on understanding the person holistically and encouraging autonomy through promoting self-management of NSSI within community settings.”

Page 11, Line 488 – “This suggests that intent influenced the type and immediacy of care delivered with SA receiving more immediate risk-focused intervention and NSSI receiving more planned, long-term therapeutic support.”

Page 11, Line 496 – “However, some trainees noted the difficulty and potential inaccuracy of using intent alone to determine risk, especially when severe NSSI or passive suicidality were evident. This highlights the limitations of static risk judgements based solely on intent as it overlooks the dynamic and complex nature of self-harm behaviours [56]. It could lead to underestimations or overestimations risk [57], potentially resulting in inadequate care or false reassurances [58]. Further investigation into how intent informs risk-related judgements is needed.”

  1. Another concern relates to the following passage: “Trainees’ attitudes towards treatment differed according to the intent of self-harm. Non-suicidal behaviours were seen as more amenable to psychological interventions. Yet, it was unclear whether they were being delivered in practice as there was also a greater focus on self-management.” If, as I understand, the focus of the paper is on acute hospital care, how could psychological interventions realistically be delivered in this setting? Clarification here would be helpful.

We thank the reviewer for this feedback. As noted in the paper on Page 3, line 124, our definition of “trainee healthcare practitioner” was:

“Trainee healthcare practitioner status was defined as studying on a university pro-gramme which would enable students to work in a healthcare practitioner role (e.g. trainee clinical psychologists, nursing associates, medical students), and which included a healthcare placement or experience in clinical settings as part of the programme.”

This definition does not specify where placements occurred and trainees typically rotate across multiple settings and services as part of their training; therefore, respondents may have been drawing on experiences in hospital settings, primary care, or other sectors. Importantly, our study explored trainees’ perceptions and attitudes rather than the actual delivery of interventions in practice.

Whilst trainees described non-suicidal self-harm as more amenable to psychological treatment, we cannot infer from these findings whether or how such interventions would be delivered in acute hospital settings. We have therefore clarified this in the manuscript (Discussion) to make explicit that our findings relate to perceived suitability rather than actual provision.

To further address these concerns, we have also added a brief note in the discussion, (limitations section) acknowledging that the availability and delivery of psychological interventions may differ between acute hospital care and community or primary care settings, and this is an important contextual factor when interpreting the findings.

Page 13, Line 559: “First, it can be noted that our findings reflect trainees’ perceptions of the suitability of interventions rather than evidence of their actual delivery. Because trainees may have drawn on multiple experiences across different healthcare settings, the availability and feasibility of psychological interventions could differ significantly across these contexts. This contextual distinction should be considered when interpreting the findings.”

  1. Finally, although limitations such as ethnicity are discussed, the significant gender imbalance in the sample is not addressed. With most participants being female (14/1, plus 1 non-binary), this represents a major limitation and should be explicitly acknowledged.

We thank the reviewer for this feedback. The authors have added to the limitations section within the discussion:

Page 13, line 564  – “The all British white sample from northwest England may limit transferability of results. Non-white practitioners may hold different views; however, this cannot be assumed without further research. Similarly, the gender imbalance within the sample further reduces the representativeness of the sample. If such differences are present in practice, this could influence clinical decision-making in ways not captured by this study. This is important to consider as non-white clinicians make up a portion of the workforce [90]. Moreover, exploring this would contribute towards understanding if demographic factors such as ethnicity and gender influence views, attitudes and clinical decision-making.”

this imbalance may have shaped the perspectives captured and should be addressed in future research.”

  1. Additionally, Figure 1 appears blurred and should be corrected.

Page 6, Line 218 – The authors of this paper have revised Figure 1.

Round 2

Reviewer 2 Report (New Reviewer)

Comments and Suggestions for Authors

Thanks to the authors for replying to my revisions. I have no further objection towards publication.

This manuscript is a resubmission of an earlier submission. The following is a list of the peer review reports and author responses from that submission.

Round 1

Reviewer 1 Report

Comments and Suggestions for Authors

This study presents the findings of analysis of interviews with 16 trainee healthcare workers around working with patients presenting with suicidal and non suicidal self harm. These are highly stigmatised bahaviours, and healthcare professionals have an opportunity to provide important care to these individuals, but frequently treat them poorly.  The authors did an excellent job of evidencing this in the introduction, my only minor comment on this section is that the rates used (6.4% and 20%) are from 2016, I wonder if these are more conservative rates than might be observed now.
Aims are well reasoned and expressed.
Methods
In the methods, there is a discrepancy between N reported in abstract (16) and N reported in demographics (15), the authors should clarify which is correct, or state if one participant declined to provide demographics. All who reported ethnicity were white, rather than 'predominantly', this will be important in considering transferability (this is noted in limitations, but it's worth reflecting on what this means in practice - how might readers use this to consider biases? how did the authors?). 
The methods also state that a sample size of 20 was deemed appropriate, but only 15/16 were collected -could the authors comment on what this means for the data and interpretation? 
Author positionality is provided for author 1, what about authors 2 and 3? what were their roles and what positionality do they bring to the interpretation and paper?
Around reflexivity and rigour, could the authors comment a little more around how they documented the progression of thinking around theme development, perhaps with examples e.g. how were codes combined to themes; what processes were used for coding - NVIVO, word, paper pen etc; what kinds of discrepancies were resolved in supervision; what did the authors do during each phase of RTA; How did they manage record keeping of reflections and decision making processes?
It would be good to provide either the interview schedule or sample questions as well to support the methodology and replicability or extension of these findings.
Results
very minor - line 228, what are some examples of 'low level' services?
In themes: Paradoxes in Perceived Motivation and Treatment Prioritization didn't really feel like the right title for what is described, which was more around trainees expressing preference for working with NSSI over SA, but prioritizing SA due to potential lethality, is that right? Perhaps clarify what the paradox is, or simplify the heading?
Discussion
Differential care is perhaps a vague term - NSSI and SA require different treatment strategies, do the authors mean less vs more empathic care? quicker vs slower? could they clarify here.
Given over half of the sample had lived experience of self-harm, it might be worth unpacking differences in responses for participants with and without lived experience to understand better how this might impact their attitudes. For example "While some trainees expressed greater optimism and motivation to treat
cases of NSSI, they often deprioritised cases when suicidal intent was not explicitly evident. This pattern reflects existing literature" - was the pattern the same for people with and without LE?
Minor - 'calculated' when applied to SA has a bit of a negative connotation, like it's manipulative, would 'purposeful' be an appropriate alternative?
These two suggestions seem to be in direct opposition: "Increasing awareness of self-harm guidelines could ensure consistent adherence in practice. [92]. Adopting a dynamic, individualised, needs-based approach [93] using risk formulations [94] may ensure intent is accounted for." Wouldn't consistent adherence to an intent-neutral framework prohibit dynamic individualised approaches? Which do the authors recommend?
line 533 - Could the authors describe Schwartz rounds
line 558 - X.X should be updated with author initials.
